# Evaluation of the Submaximal Treadmill-Based Fitness Test in Six Brachycephalic Breeds—A Follow-Up Study

**DOI:** 10.3390/ani13213413

**Published:** 2023-11-03

**Authors:** Jana D. Türkcü, Sebastian Meller, Pia S. Wiegel, Ingo Nolte, Holger A. Volk

**Affiliations:** Department of Small Animal Medicine and Surgery, University of Veterinary Medicine Hannover, 30559 Hannover, Germany; jana.denise.tuerkcue@tiho-hannover.de (J.D.T.); sebastian.meller@tiho-hannover.de (S.M.); pia.wiegel@gmx.de (P.S.W.); ingo.nolte.ir@tiho-hannover.de (I.N.)

**Keywords:** brachycephalic obstructive airway syndrome, brachycephaly, dogs, treadmill, fitness test, exercise

## Abstract

**Simple Summary:**

The application of a standardised submaximal fitness test on a treadmill has shown promising feasibility and efficacy in identifying clinical signs of brachycephalic obstructive airway syndrome in Pugs. Although early diagnosis of this disease is necessary and exercise testing combined with functional assessment has been shown to be useful, there is no evaluation of the different brachycephalic breeds. In this study, 79 dogs of different brachycephalic breeds were included to complete the fitness test under standardised conditions. Most of the examined dogs quickly adapted to running on a treadmill, so they were well monitored and consistently exercised during the test. The trotting speed can be adjusted individually, so the test also showed good applicability with other breeds. The breathing noises that occurred partially worsened over time, and some dogs only showed clinical signs of being affected until the test duration progressed. Thus, this fitness test opens the possibility of identifying affected dogs of other breeds, even if the clinical signs are only shown under exercise.

**Abstract:**

Brachycephalic obstructive airway syndrome (BOAS) in dogs challenges veterinary surgeons both with a complex clinical picture as well as wide-ranging ways to diagnose the disease, often not easily implemented nor standardised in clinical practice. The assessment of a combination of exercise testing, the occurrence of breathing noises, recovery time, and respiratory effort proved to be an appropriate method to identify Pugs with BOAS. The purpose of this study was to apply an established standardised, submaximal, treadmill-based fitness test for Pugs to other brachycephalic dog breeds. A total of 79 participants, belonging to 6 different brachycephalic breeds, trotted 15 min with an individual comfort speed of 3–7 km/h on a treadmill. Additionally, functional BOAS grading based on respiratory clinical signs before and after exercise was applied. The test was passed if the dogs presented with a BOAS grade of 0 or 1 and their vital parameters recovered to baseline within 15 min after exercise. A total of 68% showed a BOAS grade of 0 or 1 and passed the fitness test. Of the failed participants, 65% failed due to BOAS affectedness, 9% were categorised as not affected by BOAS and failed due to not passing the fitness test only, and 26% showed both failure criteria. The fitness test can be a useful method to identify BOAS-affected dogs in other brachycephalic breeds and to diagnose BOAS in dogs that only show clinical signs under exercise.

## 1. Introduction

The definition of brachycephaly and the diagnosis of brachycephalic obstructive airway syndrome (BOAS) in dogs are highly discussed current topics [1]. There is no definitive list of brachycephalic breeds because there is variation in how “brachycephaly” is defined (e.g., using skull width-to-length ratio, craniofacial ratio, craniofacial angle, etc., exclusively) [2,3]. Based on variations within the breeds, it is difficult to evaluate phenotypic characteristics as the only critical features for defining brachycephaly. Brachycephalic dogs continue to have high popularity, and the rising demand for hyper-types leads to new problems [4,5,6]. While medical treatment and surgery may provide relief for individual dogs, the only viable solutions at the population level are preventative measures through selective breeding and revision of breeding standards [5,7]. To influence effective treatments and reduce the prevalence of respective disorders, awareness among owners and the public is more necessary than ever to reduce the devastating impact on the quality of life of these dogs [2]. In addition, the public demand for more ethical and healthier breeding standards for brachycephalic breeds has risen [1,8].

BOAS is characterised by clinical signs such as stertor, stridor, inspiratory dyspnoea, increased respiratory effort, exercise intolerance, chronic shortness of breath, and a predisposition to hyperthermia [9,10,11]. In severe cases, these clinical signs may lead to decreased oxygen levels in the blood, eventual collapse, and potentially death [3,12,13]. Short-headedness and stenosis of the nares, stenosis of the nasal vestibule, obstruction of the nasal passages with hypertrophic and dysplastic bones, obstruction of the nasal passage, excessive length, massive increase in thickness of the soft palate, instability of the larynx, and other possible clinical factors must be considered [5,7,14,15].

In various studies, anatomical factors such as the degree of opening of the nares or the craniofacial ratio have shown limited significance and have not allowed a reliable statement about the presence of BOAS [5,16]. Furthermore, dog breeds exhibit different phenotypes. As each brachycephalic breed shows different expressions of brachycephaly, the effects cannot be determined using an entirely visual, external assessment [2,17,18]. An exclusively phenotypic evaluation may lead to an underestimation of the severity of BOAS expression or an incorrect diagnosis [19]. Many organisations and breeding clubs have each introduced their own exercise tests to reduce the prevalence of BOAS through more selective breeding, but these are not yet standardized and are performed under unequal conditions.

The Whole-Body Barometric Plethysmography was set as a benchmark for the characterisation of BOAS, as it is used in human medicine for the diagnosis of respiratory diseases of the upper respiratory tract and has also shown good results in respiratory disorders in cats and dogs [20,21,22]. It is a non-invasive technique for measuring respiratory function that has been initially employed for pharmacological studies in animals [20,21]. In this context, the BOAS functional grading for French Bulldogs, English Bulldogs, and Pugs was established. It classifies the severity of the disease into four different grades based on pharyngolaryngeal auscultation before and after 3 min of exercise [21]. Whole-Body Barometric Plethysmography shows excellent discrimination accuracy between the BOAS grades but requires a special experimental setup that is not available at many locations. During the validation of functional BOAS grading, in which Whole-Body Barometric Plethysmography was used as an objective comparison, a high sensitivity was achieved [23]. However, this was evaluated for three breeds of dogs, exclusively. Different types of exercise and walking tests were developed and evaluated in previous studies [6,8,16,20,24,25,26,27]. For example, the 6 min walk test and 1000 m walk test were shown to be useful in assessing BOAS severity in English Bulldogs [25]. It has been revealed that the assessment of physical performance with a 3 min trot test and a 5 min walk test is crucial for the evaluation of BOAS [21,23]. Furthermore, the potential of exercise tests in combination with pharyngolaryngeal auscultation was also established [5,23]. This suggests that exercise tests play an important role in the overall and functional assessment of BOAS diagnosis. But the physical condition of a dog is influenced to some extent by changes in factors like temperature and humidity [20]; therefore, a standardised exercise test with reproducible external conditions is crucial. In addition, the dogs may show limitations in exercise tolerance, which occur after several minutes of exercise and are shown independently of breathing noises. Thus, an extended exercise duration has proven to be beneficial. In a previous study, a standardised, treadmill-based, submaximal exercise test was evaluated for Pugs and showed promising results for functional BOAS grading [16]. The developed standardised fitness test for Pugs was easy to apply and achieved good results; therefore, the aim of the current study was to evaluate its efficiency as a diagnostic tool for other brachycephalic dog breeds. This involved testing whether the evaluation can be applied to different dog breeds, as there are differences between breeds in terms of physical and behavioural traits that may affect the feasibility of the fitness test. These differences include temperament, willingness to perform, and strength of character, among others, in addition to physical appearance.

## 2. Materials and Methods

### 2.1. Study Population and Body Characteristics

This prospective clinical study was performed at the Department of Small Animal Medicine & Surgery of the University of Veterinary Medicine Hannover, Germany, between August 2022 and May 2023. The inclusion criteria were dogs of brachycephalic breeds, which had to be adults with a minimum age of two years, as the BOAS grade usually deteriorates after the age of two years [8]. Dogs that were being treated for BOAS and had upper airway surgery were excluded. Furthermore, they had to have an unremarkable gait and physical examination, apart from breathing noises. All dogs were privately owned and acquired through social media and calls for participation by the German Kennel Association (Verband für das deutsche Hundewesen, VDH, Dortmund, Germany). In total, 79 dogs participated and were divided into 3 groups. The first and largest group consisted of 33 French Bulldogs, the second of 29 Boston Terriers, and the third of 17 dogs of the breeds Japanese Chin, Pekingese, Affenpinscher, and Brussels Griffon. 

The dogs’ chip identification data were documented. A history was taken. The bodyweight was measured, and the dogs’ constitution was graded with a body condition score (BCS) of 1–9 [28], followed by a general physical examination, including auscultation of the heart and lungs. The degree of stenosis of the nostrils was documented on the clinical examination form and via photography. The classification was based on the grading of Liu et al., which was used within the framework of the study to identify breed-specific external conformational characteristics that are associated with BOAS [29]. It categorises the stenosis via adspection into four grades: open, mild stenosis, moderate stenosis, and severe stenosis. Moreover, breathing noises at rest and after exercise were collected with a 3M Littmann electronic stethoscope (model 3100BK27, 3M Company, St. Paul, MN, USA). 

### 2.2. Submaximal Fitness Test

The study design was identical to the established fitness test for Pugs in a preceding study from the Department of Small Animal Medicine & Surgery [16]. To assess the dogs’ physical fitness, the submaximal exercise test was conducted on a motorised treadmill (“quasar”, h/p/cosmos sports and medical GmbH, Nussdorf-Traunstein, Germany). The room conditions were controlled to maintain a temperature of 20–24 °C and humidity between 30 and 60%. Before the actual fitness test, the dogs were gradually familiarised with running on the treadmill. For running on the treadmill, the dog was fitted with a harness and a leash. At the beginning, the dogs started to walk slowly on the treadmill, which was practiced a few times to give them an initial feel for the surface of the treadmill. The owner sat in front of the dog and motivated the dog with food and verbal encouragement. Gradually, the treadmill speed was increased to a level at which the dog could trot easily and steadily (about 3–7 km/h). Once this comfortable speed was found, the dog trotted for about a minute at a time to test the dog’s behaviour and willingness to move adequately for the required time. After acclimation, a 15 min break was provided [16].

The exercise test consisted of 15 min of trotting at the dog’s individual comfort speed, interrupted by 2 one-minute measurement breaks, which resulted in a total time of 17 min [16]. The values of the parameters heart rate (HR), rectal body temperature (temperature), and respiratory rate (RR) were collected at different time points (Table 1). Via pharyngolaryngeal auscultation during the measurement breaks, breathing noises were determined and categorised (see Section 2.3. Functional BOAS Grading and Breathing Noises for more details).

The test had to be stopped if the HR exceeded 220 bpm or if the workload was deemed too high to ensure that a submaximal load was not exceeded [16,30]. After the test, the dogs rested for 15 min. During this period, HR was monitored every 5 min to determine when it returned to baseline levels. RR was also monitored by counting the respiratory rate per minute. If the levels of HR and RR did not return to resting values in 15 min, with a tolerance of 10%, they were re-evaluated every 2 min. The temperature had to decrease to a physiological level of a maximum of 39.2 °C at the end of the resting phase [31]. If this was not achieved, further re-evaluations every 2 min were also performed. On the rare occasion that vital parameters reached a level that endangered health, procedures were put in place so that the emergency and critical care services would hospitalise the patient.

To be able to monitor the heart rate continuously, a Polar heart rate sensor (Polar H1 sensor and Polar FT7N monitor, Polar Electro GmbH Deutschland, Büttelborn, Germany) was attached to the dogs. As it is a submaximal workload, the heart rate should increase by about 40% compared to the resting value [16]. To measure the body temperature, a digital thermometer was used for rectal measurements.

### 2.3. Functional BOAS Grading and Breathing Noises

As in the prior and other studies, the functional grading system of Liu et al. was used, which was developed and validated in 2015 and has been used in multiple studies since then [8,16,21,25,26]. Mach et al. modified the time points of application of this system to match the exercise protocol of the study design, which was also used in the current follow-up study [16]. Each dog was classified based on an assessment of breathing noises (BN) via pharyngolaryngeal auscultation, an evaluation of the inspiratory effort, and signs of dyspnoea or cyanosis before and after 5, 10, and 15 min of exercise. According to this, there was a classification between no signs (grade 0), mild (grade 1), moderate (grade 2), or severe (grade 3) signs of BOAS. To compare the functionally impaired and unimpaired dogs, they were divided into two groups. Dogs that presented no or mild signs of BOAS (grades 0 and 1) were combined into BOAS−. The other dogs with moderate and severe signs (grades 2 and 3) were summarised in the group BOAS+. The assessment is based on a combination of BOAS grading and the exercise test. Dogs that belonged to the BOAS− category thus passed the test if they additionally showed no limitations in endurance. Hence, dogs that failed to endure or did not recover from exercise as specified failed the test, regardless of a BOAS grading. Vice versa, dogs classified as BOAS+ failed the test, regardless of passing the exercise requirements, because of their moderate to severe BOAS symptoms.

During the test, BN were differentiated between audibility with and without a stethoscope. A distinction was made between stridor and stertor and intermittent and constant. The noises that were not clear were categorised as “not assignable”. The intensity was classified into no noise (0), low-grade (1), moderate-grade (2), and high-grade (3). To determine whether the BN increased under load, the dogs were auscultated before the fitness test, twice during the test (after 5 and 10 min) and immediately after the test [16]. 

The inspiratory effort was also described and characterised as absent, mild, moderate, or severe. The presence of dyspnoea was assessed as absent, mild (signs of discomfort), moderate (irregular breathing), or severe (irregular breathing with clear signs of discomfort).

### 2.4. Statistical Design

Power analysis to determine the required number of dogs was conducted via G*Power software Version 3.1.9.6. (Heinrich Heine University Düsseldorf, Germany) [32] based on the previous study [16]. This revealed that a group size of 20 dogs per group was necessary. 

All statistical analyses were performed using GraphPad Prism Version 9.5.1. Descriptive statistics are presented as mean ± standard deviation (SD) for continuous variables or as median and range for categorical variables. As not all the data were normally distributed, mainly non-parametric statistical tests were used where required. The Shapiro–Wilk test and Kolmogorov–Smirnov tests were used to assess whether the data were normally distributed. To compare the influence of BOAS grade on HR and temperature during the exercise test and recovery period among each grade and within each breed group, a two-way ANOVA was performed with the Dunnett test as a post hoc test. To evaluate the parameters BCS and stenosis of the nostrils among the individual BOAS groups, the Kruskal–Wallis test and subsequent Dunn’s multiple comparisons tests were used. For comparison between the groups “passed” and “not passed” in relation to the parameters BCS and stenosis of the nostrils, Mann–Whitney tests were performed. Spearman’s rank correlations between stenosis of the nostrils and BCS versus the BOAS groups were used, respectively, to examine factors that depend on the individual dogs and influence the BOAS groups. *p*-values ≤ 0.05 were accepted as an indication of significance.

## 3. Results

### 3.1. Study Population

A total of 28 Boston Terriers, 32 French Bulldogs and 12 dogs of other brachycephalic breeds (4 Affenpinschers, 3 Brussels Griffons, 3 Japan Chins, and 2 Pekingeses) completed the submaximal fitness test. One French Bulldog, one Boston Terrier, one Japanese Chin, two Brussels Griffons, and two Affenpinschers did not adapt to running on the treadmill and were excluded from this study. Details of gender, age, weight, BCS distribution, and speed during the fitness test can be found in Appendix A.

### 3.2. Submaximal Fitness Test

In total, 49 of 72 dogs passed the exercise test. Of the 23 dogs that failed, 15 dogs did not pass because they were clinically affected by BOAS (BOAS+). Of these, 12 dogs were assigned to BOAS grades 2 and 3 dogs to BOAS grade 3. Furthermore, two dogs, being BOAS−, failed the test because they did not reach their initial vital parameters within recovery time. One of them was classified as BOAS grade 0, and the other as grade 1. Both impairments were detected in 6 dogs, of which 2 dogs belonged to BOAS grade 2 and 4 dogs to grade 3. Results showed that 96% of Boston Terriers, 41% of French Bulldogs, and 75% of small brachycephalic breeds passed the exercise test and presented no to mild BOAS symptoms. The fitness test was not stopped in any dog due to potential cyanosis, dyspnoea or a heart rate above 220 beats. The differences within the breeds related to the test results can be seen in Table 2. Before the test started, 8 of the 72 dogs were already panting. After 5 min, the number of panting dogs increased to 29, and after the exercise test, 49 dogs were panting. Of the dogs that panted at rest, 63% (5/8) did not pass the test. In further stages of the test, the ratio was more balanced, with a 48% (14/29) failure rate of dogs panting after 5 min. Regarding the panting dogs at the end of the test, 43% of them (21/49) did not pass. 

#### Recovery Period

If only the parameters HR and RR are considered to define the time of recovery, 36% of all dogs recovered within 5 min, 31% within 10 min, and 26% within 15 min. A total of 7% of all dogs did not recover during the 15 min recovery period. Only one dog did not reach its initial physiological body temperature during the recovery period. 

Considering the individual breeds and BOAS groups, Boston Terriers (*n* = 28) with a BOAS grade of 0 (19/28) had a decrease in HR to baseline in 5 min. One Boston Terrier required 10 min. In terms of RR, 10 dogs recovered within 5 min, 7 in 10 min, and 2 in 15 min. Of the Boston Terriers classified as grade 1 (8/28), HR recovered within 5 min in six dogs and within 10 min in two dogs. RR decreased to baseline in four dogs during the first 5 min and in two dogs each after 10 and 15 min. The Boston Terrier with grade 2 (1/28) recovered in terms of both HR and RR in the first 5 min. No dog required more than 15 min to reach a physiological body temperature.

The HR of French Bulldogs (*n* = 32) with a BOAS grade of 0 (10/32) recovered within 5 min in seven dogs and in 10 min in three. The initial RR was reached by four dogs after 5 min, four other dogs needed 10 min, and one needed 15 min. One dog still had an increased RR after 15 min. Within the group of dogs classified as grade 1 (5/32), four dogs required 5 min to lower the RR, and one dog required 15 min. The RR decreased to baseline in 5 min in four dogs and in 10 min in one dog. In this group, one dog was unable to regulate its temperature to a physiological level within 15 min. Of the dogs that were classified as grade 2 (10/32), the HR of nine dogs decreased to baseline levels within 5 min and in one dog within 10 min. With regard to RR, three dogs each recovered after 5, 10, and 15 min; moreover, one dog did not recover within the 15 min. In the group of dogs with grade 3 (7/32) BOAS, HR decreased in five dogs in 5 min and in one dog each after 10 and 15 min. The RR was recovered in three dogs in 5 min and in two after 15 min. In another two dogs, recovery could not occur within the time of 15 min.

Considering the small brachycephalic breeds (*n* = 12), it was observed that the HR of all dogs in the BOAS grade 0 group (5/12) recovered in 5 min. In contrast, the RR decreased in all of them after 15 min. In the dogs with a BOAS grade of 1 (4/12), the HR decreased in two dogs each after 5 and 10 min. The RR could be reached within 5 min for three dogs, whereas one dog needed 15 min. The HR of dogs with grade 2 (3/12) recovered after 10 min in one dog and after 15 min in two dogs. However, the RR decreased after 5 min in two dogs and after 10 min in one dog. No dog of these breeds required more than 15 min of recovery for temperature.

### 3.3. Functional BOAS Grading

Since the dogs could not all be familiarised with running on a treadmill, BOAS grading was applied to 72 of the original 79 dogs. No abnormal respiratory sounds or increased inspiratory effort were detected in 34 dogs; therefore, they were assigned BOAS grade 0. 

Based on 38 dogs with a BOAS grade between 1 and 3, 10 dogs were already assigned to their final BOAS group at the beginning (BOAS Group 1: three dogs; BOAS Group 2: four dogs; and BOAS Group 3: three dogs). With a total of 14 dogs, most of the participants were correctly classified into their final group after 5 min of trotting. In this case, seven dogs were classified in BOAS Group 1, four dogs in BOAS Group 2, and three dogs in BOAS Group 3. After 11 min, 10 more dogs reached their final BOAS grade, with 5 dogs each in BOAS groups 2 and 3. At the end of the test, the BN increased in three dogs, so that one dog each moved into BOAS groups 1, 2, and 3. Table 3 shows the distribution of dogs in the individual BOAS groups at the different measurement time points.

#### Breathing Noises

At rest, before the test started, 53 of 72 dogs showed no BN. After 5 min of trotting, the number of unaffected dogs decreased to 43, and after 11 min, to 37 dogs. When the dogs were auscultated after 17 min, it was found that only 34 dogs without BN remained, and these therefore did not show a BN at any time. Mild BN increased in intensity as time progressed and often increased in frequency as well. Some dogs showed breathing sounds only at one measurement point, which is also because some dogs alternated between normal breathing and panting so that breath sounds could not be evaluated properly. Regarding the different types, it was determined that 68% were classified as stertor and 13% as stridor. In 18%, both types of noise were identified. The different grades of observed BN in all dogs can be seen in Table 4. The occurrence of BN subdivided by breeds can be seen in Appendix A.

### 3.4. Vital Parameters and BOAS Groups

#### 3.4.1. Heart Rate

In a first step, the dogs from the respective BOAS groups were combined, and the documented HR values were compared once overall at each time point (Figure 1A). A two-way ANOVA showed a significant effect of BOAS grade and time interaction on HRs (*p* = 0.0128). After applying Dunnett’s multiple comparisons test, significant differences between BOAS group 0 and the other groups were found, especially during breaks and recovery (see Figure 1A for more details). 

Subsequently, the BOAS groups within the respective breed groups were analysed with a two-way ANOVA. In the Boston Terriers, there were no significant effects on the HRs. Due to the small number of animals in BOAS group 2 (*n* = 1), only BOAS groups 0 and 1 were considered for comparison (Figure 1B). In the French Bulldogs, there was an effect of BOAS grouping on HRs (*p* = 0.0126). Dunnett’s multiple comparisons testing revealed significant differences between the BOAS 0 and the other groups during breaks and recovery periods (see Figure 1C for more details). When analysing the small brachycephalic dog breeds, significant differences between BOAS groups 0 and 2 were mainly observed during the recovery phase (see Figure 1D for more details). 

#### 3.4.2. Temperature

The evaluation of whether the BOAS degree had an impact on the body temperature during the exercise test and recovery is shown in Figure 2. When analysing all dogs, a significant difference was found only after the exercise period between BOAS groups 0 and 2 (*p* = 0.0428). No significant effects or differences were observed when analysing breed groups. See Figure 2 for further details.

#### 3.4.3. Respiratory Rate

In terms of RR, it was noticeable that BOAS-positive dogs started panting earlier and panted longer during the recovery period. Specifically, at the end of the test at 17:00 min, 59% of the dogs in BOAS groups 0 and 1 were panting, whereas in contrast, 86% of the dogs in BOAS Group 2 and 100% of the dogs in BOAS group 3 were panting. The exact percentages at the measuring points can be found in Table 5.

In the same course, the inspiratory effort was recorded. Before the fitness test started, 10 of 72 dogs showed a mild inspiratory effort, with an increase to 45 dogs after 5 min. Another 5 min later, 57 dogs showed mild and 2 showed moderate inspiratory effort. Immediately after the test, a moderate inspiratory effort was detected in 5 dogs and a mild one in 59 dogs. 

### 3.5. Body Characteristics, BOAS Groups, and Test Results

During the clinical examination, BCS (Scores 1–9) and grade of stenosis of the nostrils were assessed (Scores 1–4). A total of 23 of the dogs were overweight with a BCS > 5. On visual inspection of the nares, it was observed that 14% were open (Score 1), 46% had a mild stenosis (Score 2), 31% had a moderate stenosis (Score 3), and 10% had a severe stenosis (Score 4). The correlation between BCS and BOAS grades showed a medium correlation (r = 0.36; *p* = 0.0009). The correlation between the stenosis of the nares and BOAS grades also showed a moderate correlation (r = 0.4106; *p* = 0.0003). 

Further statistical tests also indicated a relationship between the stenosis of the nostrils and the BOAS grade (*p* = 0.0039). In particular, a significant difference was observed in BOAS group 0 versus BOAS group 3 (*p* = 0.0023) (Figure 3A). The analysis of the influence of the BCS on the BOAS groups also revealed a significant effect (*p* = 0.0127). More precisely, a significant difference was only found between BOAS group 0 and BOAS group 2 (*p* = 0.0185) (Figure 3B). 

Finally, the influence of the parameters stenosis of the nostrils and BCS on the fitness test results was investigated, respectively (Figure 3C,D). A significant difference was found between the test results in relation to stenosis (*p* = 0.0008) as well as between the test results in relation to BCS (*p* = 0.0004).

## 4. Discussion

In this study, the feasibility and applicability of the submaximal treadmill-based fitness test developed for Pugs were evaluated in other brachycephalic breeds. Since BOAS is a multimodal disease and the influence of individual anatomical abnormalities on the clinical signs observed in individual animals may vary [25,33], exercise tests are used as a reliable method for diagnosing the disease. Still, they have only been tested in a limited number of brachycephalic breeds [16,21]. The submaximal fitness test evaluated in this study was well applicable to all selected dog breeds and could be easily performed with the equipment commonly available in many clinics. One of the advantages of running on a treadmill is that dogs can complete the test in a structured procedure with limited influences from environmental or climatic effects [34]. Vital signs can be monitored continuously so that physiological responses can be analysed and any workload-related problems can be handled immediately. Most dogs were quickly adapted to running on the treadmill, as only around one in eleven dogs could not get used to running in this environment. Part of the reason for this could be that many owners reported that their dogs very rarely, if ever, walked with a harness. Similar to the first study, a comparable number of dogs could be acclimatised to running on the treadmill. Similarly, other studies showed that the treadmill is not tolerated by only a small proportion of dogs without prior experience [6,16,35,36,37]. A constant, controlled workload of the dogs can thus be achieved, and submaximal exercise tests are easy to perform [25]. As the trotting speed is determined individually for each dog within the familiarisation period, it is possible to test different breeds using the same exercise protocol. 

BOAS is a progressive disease, with the first clinical signs usually appearing between 2 and 4 years and worsening with age [10,38]. For this reason, a minimum age of 2 years was set as a prerequisite for participation in the study. This age would also be recommended for breeding selection [8]. It must be considered that the exercise test is only a reflection of the current state of health and cannot be used to predict future affection [8]. It is accordingly recommended to repeat the test at regular intervals, for example, annually until four years of age, in order to diagnose affected animals. 

To assess the severity of BOAS, a grading system for clinical evaluation by Liu et al. [21] was used. This and modifications thereof had already been shown to be well applicable and meaningful in other studies [8,16,19,23,25,26] and this was confirmed as well. However, the evaluation of the different grades by a veterinarian is always associated with a certain subjectivity. 

The previously designed exercise protocol by Mach et al. [16] showed good applicability, and the time of 15 min of exercise proved to be necessary and appropriate. The BOAS grading, as described above, was developed based on an exercise duration of 3 min at trot speed [21]. Most of the dogs were classified into their BOAS group after 5 min. Still, it proved useful to extend the duration of the test, since after 10 and 15 min, an additional 13 of 38 dogs were classified with BOAS signs. 

A 40% increase in HR compared to resting values indicates that the dog is being exercised submaximal [36], so measurement at different time points is necessary. Some dogs are very nervous, especially before the test, due to the unfamiliar environment. Therefore, further analysis of the HR level in an individual dog is not meaningful. Even though the standardised conditions can exclude factors like temperature, there are still individual factors such as the condition of the dog’s general fitness, anxiety, etc., that make it difficult to compare individual values. 

A comparison of the mean HRs of all dogs classified into BOAS groups revealed significant differences between BOAS groups 0 and 2 and BOAS groups 0 and 3, especially in the recovery phase and during the breaks. As the HR during these time points did not decrease as much as the HR of the dogs not being affected, it can be assumed that the fitness of dogs with BOAS grades 2 and 3 may be impaired. The HR of the BOAS-affected dogs was not higher at all time points, but it is noticeable that the BOAS-unaffected dogs showed a faster recovery where the heart rate dropped below the pre-test baseline. HR recovery is a meaningful parameter, as it has even been shown to be a prognostic tool for all-cause mortality [39]. Therefore, the extended recovery time could indicate limited resilience. It was also apparent that the small dogs generally had a higher heart rate than the other groups. This is, of course, due to anatomical and physiological reasons, but during the examination, it was noticeable that the dogs were significantly more anxious than, for example, most of the French Bulldogs. To determine actual fitness status, other parameters would need to be collected, such as VO2max, the maximum capacity for oxygen consumption by the body during exercise, which is commonly used in horses and humans to determine aerobic and cardiovascular fitness [30]. 

When analysing temperature before, immediately after the test, and after the recovery period, only one statistically significant difference was shown at the end of exercise between BOAS groups 0 and 2. The mean differences within the different breeds were only marginal. However, when considering the graphs in Figure 2, it is evident that the temperature of the BOAS-positive dogs is significantly higher than that of the BOAS-negative dogs. Again, it was noticeable that the temperature of the small breeds was much higher than in the other breed groups. The temperature of the Boston Terrier in BOAS group 2 was also higher, but this was a single individual, so no conclusions can be derived from this. In the previous study comparing the Pug group with the control group (10 healthy mesocephalic dogs), there was evidence from statistically significantly higher temperature values that heat exchange is more impaired in dogs with clinical signs of BOAS [16]. Other studies have shown an increase in temperature per BOAS grade [25] and significant differences between mesocephalic and brachycephalic dogs [16,19,25,26]. Dogs are geared to dissipate heat through the nasal conchae, so they depend on nasal breathing for thermoregulation [7,25,33]. Anatomical changes such as abnormally growing turbinates and stenotic cartilages in brachycephalic breeds can lead to inadequate thermoregulation and heat intolerance [11,17,26].

Increased RR was generally expected due to higher oxygen demand during exercise [6], and it was noticeable that most dogs were affected by BOAS panting earlier throughout the test period. Panting is the primary way of evaporative cooling in dogs exposed to heat or increased physical activity [40]. Inhalation through the nose and exhalation through the mouth cause evaporation to occur over the large surface area of the nasal mucosa [38]. Aberrant and malformed conchal tissue, among other factors, obstructs the nasal airways of brachycephalic dogs to varying degrees and increases airway resistance [3]. In addition, the nostrils of dogs with stenotic nares are usually immobile; therefore, they cannot abduct them, which also increases airway resistance [41,42]. The surface area of the nasal mucosa is reduced, and the lumen of the nasal passages narrows, which restricts the ability of affected dogs to thermoregulate [7,14]. These limitations of the dogs explain the increased panting of BOAS-affected dogs during and after the exercise test compared to non-BOAS-affected dogs.

The classification of BN revealed more than two-thirds to be stertors and only 13.2% to be stridors. Stertor, a lower-pitched, snoring sound, is associated with nasal, pharyngeal, and tracheal diseases, e.g., excessive tissue in the upper portion of the airway, such as an elongated soft palate [12,43]. On the contrary, stridor is associated with laryngeal and extra-thoracic tracheal disorders and is characterised by a high-pitched, whistling sound [23,42]. Since an elongated soft palate is the most common abnormality, the stertor is the most frequently observed noise here [12,15].

The duration of the recovery period is necessary to determine how quickly the dog can recover from the submaximal workload. To include this in the evaluation, as it is an important part, a maximum time of 15 min was set. This was also shown to be an appropriate duration in other studies [6,8,16,26]. If the time for recovery had been reduced to 10 min, 67% of the dogs would have passed. The initial HR was recovered the fastest, while the RR required a longer time. One dog could not reach physiological temperature during the recovery period. 

As obesity is a predictor of increased BOAS risk [44], the BCS was assessed during a clinical examination. Obesity can lead to an accumulation of fat tissue in the palate, tongue, and tissues around the airways, narrowing them further [44]. A total of 32% of the presented dogs were overweight. Since the incidence of overweight dogs is about 20–40% [45], no statement can be made about the increased overweight of brachycephalic breeds. Comparing the BOAS groups with regard to BCS, it can be shown that there is a correlation between the severity of BOAS and the level of BCS, which confirms the aforementioned findings of the current study. Moreover, a relationship between BCS and the result of the fitness test was determined. Weight loss also reduces the accumulation of fatty tissue in soft tissue structures in the head and neck, which should alleviate airway constriction. Especially for mildly affected individuals, weight loss can therefore be helpful in reducing BOAS severity. Severely BOAS-affected dogs often exhibit distinct secondary changes, and while weight loss can be helpful, it might not be sufficient as a standalone treatment measure. As an additional assessment, the dogs could be reevaluated with the fitness test after weight loss to evaluate if the BOAS score decreased and to what extent the weight loss supports the dogs’ well-being. Another conformational risk factor is the stenosis of the nostrils. The subjective evaluation of the stenosis using a 4-point scale [29] proved to be quick and easy to use both here and in advance [16,26]. The stenosis of the nostrils also showed a significant influence on the BOAS grade as well as on the fitness test result. It should be taken into account that it is possible that some dogs do not have a moderate- or high-grade nasal stenosis but other respiratory abnormalities [7]. In addition to the shortness of the head and the other anatomical characteristics that are visible externally, it is important to remember that most malformations are located behind the nostrils [10,46,47]. It is not possible to determine via nasal evaluation solely whether additional intranasal stenosis or other malformations are present that limit the welfare of the dog. Hence, a functional examination of the dogs is very important.

This study has limitations in several aspects. The first limitation is the verifiability of the participation requirements. Dogs that have already undergone surgery could still be presented and thus pass the test. It is difficult to determine based on an external assessment whether the dogs have already been operated on. However, this problem is present in all diagnostic methods related to BOAS. Since the breed groups are too small to make a breed-specific statement about the population, the test would have to be applied to further dogs and evaluated. The search for test participants proved to be difficult for the small brachycephalic breeds, as the litter numbers have decreased over the years. The Affenpinscher recorded 23 puppies in a statistic of the *Verband für das deutsche Hundewesen* (VDH, German kennel club), in 2019 [48]. In 2021, only seven puppies were reported. The same is seen in Japan Chins and Griffons, and therefore not many participants were found. In contrast, due to the further popularity and the growing number of pets since the last few years, the number of registered French Bulldogs and Boston Terriers increased. In 2022, the French Bulldog was the fourth most popular dog breed in Germany, with 12,452 new registrations [49]. Nevertheless, in order to be able to find out tendencies between the groups, the Affenpinschers, Brussels Griffons, Japan Chins, and Pekingeses were combined in one group, as the calculated number of dogs could not be achieved for these breeds. This should be seen critically. The phenotypes vary between these breeds, and they therefore pose different breed-related problems [7]. The comparison with this group was not representative, and this should be considered when interpreting the results of this study. In addition, due to the call through the kennel club, many breeders attended the test and might only bring dogs to the test that they thought would pass. Therefore, the results of the distribution of BOAS grades can only be transferred to the population to a limited extent. Of 72 dogs, 7 dogs were found to be affected by BOAS of grade 3. In order to be able to make more detailed statements here, more dogs belonging to this group should be recruited as test participants.

Since the evaluation of the BOAS grade cannot guarantee objectivity, the assessment of severity could vary depending on the veterinary surgeon [25]. Therefore, adequate training is necessary to ensure that the dogs are evaluated as objectively as possible. Another possible approach would be to implement an automated evaluation, e.g., via artificial intelligence and machine learning, which classifies the breathing noises independently. In human and partly in veterinary medicine, such methods have already been tested, and those latest techniques have shown favourable performance in some contexts [50,51,52,53].

## 5. Conclusions

Based on this follow-up study, the standardised submaximal fitness test for Pugs performed on the treadmill has proven to be easy to implement and to be well applicable to other brachycephalic breeds. By individually adjusting the trotting speed, this test is also easily adapted to other brachycephalic breeds of different sizes. With an exercise duration of 15 min, good diagnostic results can be achieved. It is recommended to repeat the test annually during the first years of life in order to diagnose an incipient BOAS as quickly as possible and to enable the dogs to live a more comfortable life with early treatment.

## Figures and Tables

**Figure 1 animals-13-03413-f001:**
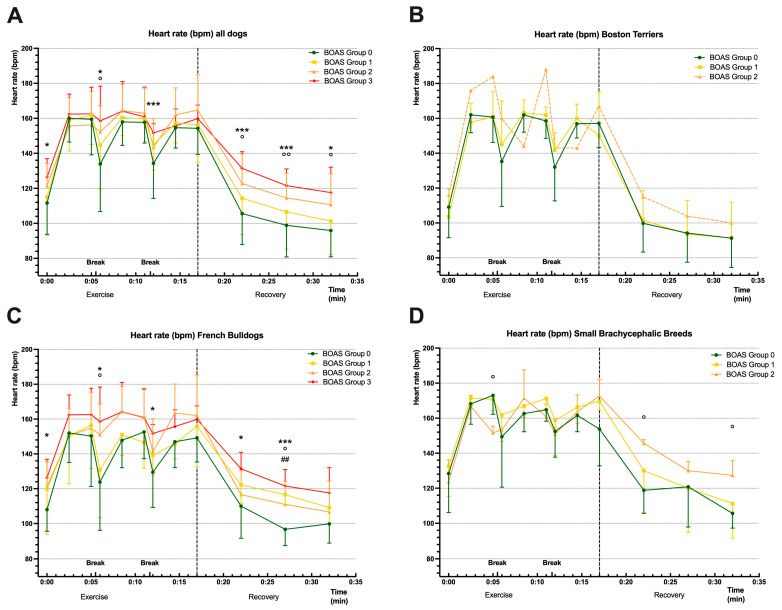
Mean ± SD of heart rate (HR; beats per minute, bpm) of all dogs ((**A**); *n* = 72), Boston Terriers ((**B**); *n* = 28), French Bulldogs ((**C**); *n* = 32), and small brachycephalic breeds ((**D**); *n* = 12), subdivided into brachycephalic obstructive airway syndrome (BOAS) groups (coloured lines). HR was recorded at rest (time point 0), during exercise, and during recovery period. The dotted vertical line separates the exercise and the recovery periods. The dotted HR line of BOAS group 2 in (**B**) is attributed to a sample size of *n* = 1 (excluded from statistical analysis). Significant differences derived from two-way ANOVA and Dunnett’s multiple comparisons are marked with hashtags (## = *p* < 0.01) for BOAS group 0 versus BOAS group 1, with circles (° = *p* < 0.05; °° = *p* < 0.01) for BOAS group 0 versus BOAS group 2, and with asterisks (* = *p* < 0.05; *** = *p* < 0.001) for BOAS group 0 versus BOAS group 3.

**Figure 2 animals-13-03413-f002:**
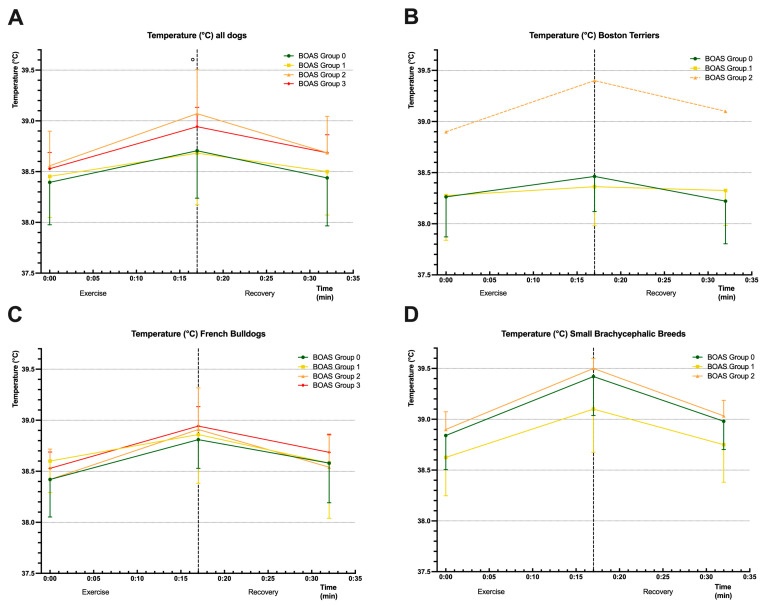
Mean ± SD body temperature (°C) of all dogs ((**A**); *n* = 72), Boston Terriers ((**B**); *n* = 28), French Bulldogs ((**C**); *n* = 32), and small brachycephalic breeds ((**D**); *n* = 12), subdivided into brachycephalic obstructive airway syndrome (BOAS) groups (coloured lines). Temperature was measured at rest (min 0), after exercise (min 17), and after recovery period (min 32). The dotted vertical line separates the exercise and the recovery periods. The dotted body temperature line of BOAS group 2 in B is attributed to a sample size of *n* = 1 (excluded from statistical analysis). A significant difference derived from two-way ANOVA and Dunnett’s multiple comparisons testing was only found between BOAS group 0 and BOAS group 2 in (**A**) at min 17, marked with a circle (° = *p* < 0.05).

**Figure 3 animals-13-03413-f003:**
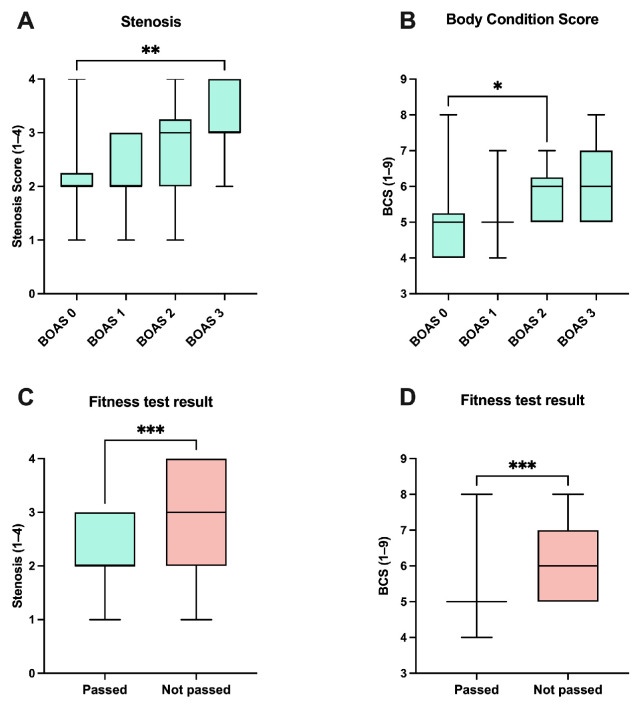
Box and whisker plots with corresponding medians are shown. (**A**) shows the stenosis of nostrils (Scores 1–4), and (**B**) shows the body condition score (BCS) (Scores 1–9) in relation to brachycephalic obstructive airway syndrome (BOAS) groups, respectively (group 0: *n* = 34; group 1: *n* = 17; group 2: *n* = 14; and group 3: *n* = 7). (**C**,**D**) show stenosis scores and BCS in relation to the fitness test results (passed, *n* = 49; not passed, *n* = 23), respectively. In (**A**,**B**), the Kruskal–Wallis test (*p* = 0.0039 and *p* = 0.0127) followed by Dunn’s multiple comparisons tests (** = *p* < 0.01 and * = *p* < 0.05) was used to assess the parameter stenosis scores and BCS between BOAS group 0 and the other groups, respectively. In (**C**,**D**), Mann–Whitney tests were used for the comparison between the “passed” and “not passed” groups for the parameter stenosis scores and BCS, respectively (*** = *p* < 0.001).

**Table 1 animals-13-03413-t001:** Measurement time points in exercise and recovery periods. Heart rate recorded in beats per minute (bpm), respiratory rate in breaths per minute (breaths/min), and rectal body temperature in degrees (°C). Pharyngolaryngeal auscultation was performed using a stethoscope. Time points at which measurements were conducted are marked with an “x”.

	Time Point	Heart Rate	Respiratory Rate	Temperature	Auscultation Incl.
	(min)	(bpm)	(Breaths/min)	(°C)	BOAS Grading
Exercise	At Rest	x	x	x	x
	02:30	x			
	05:00	x	x		x
	06:00	x			
	08:30	x			
	11:00	x	x		x
	12:00	x			
	14:30	x			
	17:00	x	x	x	x
Recovery	05:00	x	x		
	10:00	x	x		
	15:00	x	x	x	

**Table 2 animals-13-03413-t002:** Test results are divided into breed groups. The category “not passed” is subdivided into not passed because of BOAS symptoms (BOAS), failed fitness test (FT), or not passed because both impairments occurred. Those who could not be familiarised with the treadmill and were therefore excluded before the fitness test began are in the category “No familiarisation”.

Test Results	Total	Boston Terriers	French Bulldogs	Small Brachycephalic Breeds
Total	72 (100%)	28/72 (39%)	32/72 (44%)	12/72 (17%)
Passed	49/72 (68%)	27/28 (96%)	13/32 (41%)	9/12 (75%)
Not passed	23/72 (32%)	1/28 (4%)	19/32 (59%)	3/12 (25%)
Not passed (BOAS)	15/23 (65%)	1/1 (100%)	11/19 (58%)	3/3 (100%)
Not passed (FT)	2/23 (9%)		2/19 (10%)	
Not passed (BOAS + FT)	6/23 (26%)		6/19 (32%)	
No familiarisation	7/79 (9%)	1/29 (3%)	1/33 (3%)	5/17 (29%)

**Table 3 animals-13-03413-t003:** Number of total dogs classified into brachycephalic obstructive airway syndrome (BOAS) grades at different time points.

	Time Point	BOAS−		BOAS+	
BOAS Grading		Grade 0	Grade 1	Grade 2	Grade 3
	0 Min	53/72 (74%)	8/72 (11%)	8/72 (11%)	3/72 (4%)
	5 Min	43/72 (60%)	14/72 (19%)	9/72 (13%)	6/72 (8%)
	11 Min	37/72 (51%)	15/72 (21%)	14/72 (19%)	6/72 (8%)
	17 Min	34/72 (47%)	17/72 (24%)	14/72 (19%)	7/72 (10%)

**Table 4 animals-13-03413-t004:** Breathing noises (BN) of all dogs at rest and after 5, 10, and 17 min of exercise.

			All Dogs		
Time Point		At Rest	5 min	11 min	17 min
Number of subjects		72	72	72	72
No BN		53/72 (74%)	43/72 (60%)	37/72 (51%)	39/72 (54%)
	BN audible without stethoscope (*n*)				
	Mild	4	3	7	10
Intermittent	Moderate	1	2	1	0
	Severe	0	0	0	0
	Mild	1	4	5	5
Constant	Moderate	6	4	6	6
	Severe	2	4	4	5
	BN audible only with stethoscope (*n*)				
	Mild	3	5	9	6
Intermittent	Moderate	0	1	0	0
	Severe	0	0	0	0
	Mild	1	3	2	1
Constant	Moderate	1	2	1	0
	Severe	0	0	0	0

**Table 5 animals-13-03413-t005:** Number of panting dogs in %, divided by brachycephalic obstructive airway syndrome (BOAS) groups at different measurement times.

	Time Point (min)	BOAS 0 (*n* = 34)	BOAS 1 (*n* = 17)	BOAS 2 (*n* = 14)	BOAS 3 (*n* = 7)
	At Rest	3%	12%	21%	29%
	05:00	35%	24%	57%	71%
Exercise	11:00	53%	41%	57%	86%
	17:00	59%	59%	86%	100%
	05:00	32%	24%	71%	57%
Recovery	10:00	21%	12%	36%	29%
	15:00	6%	12%	21%	14%

## Data Availability

The data presented in this study are available on request from the corresponding author.

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
