# Peer review of "Evaluation of the Submaximal Treadmill-Based Fitness Test in Six Brachycephalic Breeds—A Follow-Up Study"

_animals, 2023, doi:10.3390/ani13213413_

Round 1
Reviewer 1 Report
Comments and Suggestions for Authors
Comments to the Author:
Regarding the evaluation of the manuscript entitled "Evaluation of the submaximal treadmill-based Fitness Test in 5 brachycephalic breeds – a follow-up study
· Please describe how was the sample size calculated?
· Page 2, Lines 94-96: Please clarify the point you are attempting to make. "the dogs may show limitations in exercise tolerance that do not occur within a few minutes and are shown independently of breathing noises".
· Page 15-16, Lines 472- 474: The idea behind this work is good but discussion needs precision and clarity. Could you elaborate more on this by specifically talking about "the dogs could be reevaluated with the fitness test after weight loss to see if the BOAS score decreases and the dogs are therefore less affected"
Author Response
Dear Reviewer,
Thank you very much for taking the time to review this manuscript. Please find the detailed responses below and the corresponding revisions in the re-submitted files.
- Please describe how was the sample size calculated?
- Thank you for this question. The sample size was calculated with a G power analysis. This information and the calculated group size had been added (Line 202 – 204).
- Page 2, Lines 94-96: Please clarify the point you are attempting to make. "the dogs may show limitations in exercise tolerance that do not occur within a few minutes and are shown independently of breathing noises".
- Thank you for the comment. It has been added that this indicates that an extended exercise duration is beneficial (Line 96 – 97).
- Page 15-16, Lines 472- 474: The idea behind this work is good but discussion needs precision and clarity. Could you elaborate more on this by specifically talking about "the dogs could be reevaluated with the fitness test after weight loss to see if the BOAS score decreases and the dogs are therefore less affected"
- Thank you for the comment. We appreciate that this has not been as clear. We have now clarified this in the text as requested (Line 516 - 522).
Reviewer 2 Report
Comments and Suggestions for Authors
The evaluated study proposes the adaptation in several breeds of brachycephalic dogs of a functional test for classifying the degree of BAOS that has been previously used and validated in pugs.
The study is correctly planned and uses a solid and rigorous methodology that gives full confidence that the results are firm and adequately valued and discussed. It is also a topic of great practical interest that can help objective decision-making about these patients in a clinical environment. Although there are limitations that the authors themselves conveniently expose, the positive aspects indicated are many and therefore this reviewer considers that the work deserves to be published in this journal.
Some deficiencies have been detected that are listed below and that must first be addressed before final acceptance:
GENERAL SUGGESTIONS:
- The material and methods section requires a review to clarify multiple aspects of the methodology used. Overall, it is not explained in a clear and simple way, and I believe that a general effort can still be made in that direction.
- The results section includes many tables that are not as illustrative as they should be. Numerical results can be grouped (fractions instead of “n” and percentages below in parentheses, for example). You can avoid an excess of zero results and try to display or group them in another way. The number of tables can perhaps be reduced then. In particular, putting what is in the text into a table is unnecessary or vice versa. Expressing it in text may be more practical if the tables contain many zeros.
SPECIFIC SUGGESTIONS
Line 27: Do not start a sentence with a number (review the rest of the document and correct similar ones)
Line 28: individual comfort speed: indicate the range
Line 28: a functional BOAS grading: indicate what it is based on
Line 72-97: This writing has more discussion style than introduction one. It would be better to give more direct ideas of what is known and what relevance the test has. Leave comparisons between studies for discussion.
Line 98: the sentence is a continuation of the previous paragraph, do not open a new paragraph
Line 111: dogs were being treated for BOAS: what treatments are included?
Line 113: free of general diseases: specify what you consider “general diseases”
Line 120: body condition score (BCS): include reference
Line 121-122: The degree of stenosis of the nostrils was: although the reference is included, it must be briefly specified how this gradation was made
Line 147: breathing noises were determined and categorized: must be specified below or indicate in which section of the study this information is included
Line 150: RR were monitored: indicate a method or section of the manuscript where it is specified
Line 169-170: the present study design: the specific summative criteria for this grading for the study must be specified
Line 166: This section is especially important in this study and is especially confusing as it is expressed. It requires an effort of clarity.
Line 212-224: do not duplicate information if it is already in tables
Line 242: Table 3: Percentages can be placed next to the n in parentheses and use fractions instead of just n
Line 253: many zero values. Find another way to display the results or convert it briefly to text.
Line 259: No respiratory sounds: abnormal??
Line 270: table 5: provide % values of the total and or by subgroups
Line 282: The data would be more illustrative as a fraction and % in parentheses. Abbreviations should be included in the table footer. This table is very big with many zeros. Should be summarized or deleted
Line 337: Table 7: number or %??
Line 376: Clarify in what context
Reviewer 3 Report
Comments and Suggestions for Authors
This is a well described study on the effect of 15 minutes of exercise on a treadmill at the preferred speed of brachycephalic dogs on clinical symptoms and physiologic variables of the dogs.
The paragraph 2.4 Statistical design requires revision because the sentence “since the data were not normally distributed, non-parametric statistical tests were used” implies that all data were not normally distributed. However, this seems not to have been the case, because many statistical examinations were done with parametric tests. Thus, please write that data, that were not distributed normally were examined with “X and Y” tests and the normally distributed data with “XX and YY” tests.
Please explain why it was preferred to run dogs at the speed they selected instead of testing them at standardized speeds (eventually according to their breed or size). Such would allow for comparability with other and future studies and may elicit larger (and faster) effects on clinical parameters and physiologic variables than running the dogs at their preferred speeds.
Detailed comments
- Replace “the” for “a” in the title
- Pugs is written often with small “p” but other times with large “P”. I like to suggest that it is written always with large “P”.
- Temperature is sometimes fully written, sometimes written “Temp.” and sometimes “T” only. I suggest to write “temperature” always.
- The tables and figures should be understood without having to read the text. Thus, please write all abbreviations in full.
Page 2
Line 52: Replace “However” for “In addition”
Line 86: Write MWT in full
Page 3
Line 111: Replace “In the case that dogs” for “Dogs that”
Line 112: Delete “have”
Page 5, line 216: Add “s” to Terrier
Page 8, table 6: For the first time appears the abbreviation “RN”. Does it mean “Respiratory Noise”? If so, which is the difference to “Breathing Noise, BN”? Like described above already: Tables and figures need to be self-explanatory!
Page 14, line 401: Replace “submaximal exercised” for “exercised submaximal”
Page 15
Line 432: Replace “had” for “have”
Line 461-462: The sentence “If the time had been reduced to 10 minutes, only 66.7% of the dogs would have passed.” is not conclusive. What do the authors like to explain?
Comments on the Quality of English Language
Can be read well. Minor corrections only.
Round 2
Reviewer 3 Report
Comments and Suggestions for Authors
Thank you very much for revising the statistics and descriptions of tables and figures!
My only “complaint” is that I miss your reasoning in the manuscript for not running “healthy” dogs at defined speeds to allow for standardization and therefore comparison between studies.
You wrote in your response to my queries:
In general, a submaximal exercise level must be obtained. In the previous study, it was determined that running the dogs at an individual trotting speed leads to this exercise level being achieved. For this, care was taken that the heart rate increased by at least 40% compared to the resting value. To define different speeds according to breed and size, a large number of dogs would have to be tested to determine the average speed. Especially for the small breeds. Some of the breeds were limited in numbers and very difficult to recruit.
Write this in the manuscript for all to know your thoughts on the matter. I continue to think that it would be better to standardize the speed of the 15 minutes exercise bout. By now there should be enough data to prescribe speeds for dogs of different size, if necessary. In addition, standardized speeds may elicit larger (and faster) effects on clinical parameters and physiologic variables than running the dogs at their preferred speeds which may be related to the grade of BOAS.